# Knowledge, perceptions and preventive practices towards COVID-19 early in the outbreak among Jimma university medical center visitors, Southwest Ethiopia

**Yohannes Kebede**[1]*, **Yimenu Yitayih**[2], **Zewdie Birhanu**[1], **Seblework Mekonen**[3], **Argaw Ambelu**[3]

**1** Department of Health, Behavior, and Society, Faculty of Public Health, Jimma University, Jimma, Ethiopia, **2** Psychiatry Department, Faculty of Health Science, Jimma University, Jimma, Ethiopia, **3** Department of Environmental Health Science and Technology, Faculty of Public Health, Jimma University, Jimma, Ethiopia

* yohanneskbd@gmail.com

## Abstract

### Background

Novel-coronavirus disease-2019 (COVID-19) is currently a pandemic and public health emergency of international concern, as avowed by the World Health Organization (WHO). Ethiopia has become one of the affected countries as of March 15, 2020.

### Objective

This study aimed to assess the knowledge, perceptions, and practices among the Jimma University medical center (JUMC) visitors in Jimma town.

### Methods

A cross-sectional study was conducted on 247 sampled visitors, from 20–24 March 2020. Consecutive sampling was used to recruit the participants. The study tools were adapted from WHO resources. The data were analyzed using the Statistical Package for Social Sciences (SPSS) version 20.0. Descriptive statistics were used to describe the status of knowledge, perception, and practices. Logistic regression was executed to assess the predictors of dominant preventive practices.

### Results

Of the 247 respondents, 205 (83.0%) knew the main clinical symptoms of COVID-19. 72.0% knew that older people who have chronic illnesses are at high risk of developing a severe form of COVID-19. About 95.1% knew that the COVID-19 virus spreads via respiratory droplets of infected people, while 77 (31.2%) of the respondents knew about the possibility of asymptomatic transmission. Only 15 (6.1%) knew that children and young adults had to involve preventive measures. Overall, 41.3% of the visitors had high knowledge. The majority, 170(68.8%), felt self-efficacious to controlling COVID-19. 207(83.3%) believed that

**Data Availability Statement:** All relevant data are within the manuscript and its Supporting Information files.

**Funding:** The author(s) received no specific funding for this work.

**Competing interests:** The authors have declared that no competing interests exist.

COVID-19 is a stigmatized disease. Frequent hand washing (77.3%) and avoidance of shaking hands (53.8%) were the dominant practices. Knowledge status and self-efficacy (positively), older age, and unemployment (negatively) predicted hand washing and avoidance of handshaking.

## Conclusions

The status of knowledge and desirable practices were not sufficient enough to combat this rapidly spreading virus. COVID-19 risk communication and public education efforts should focus on building an appropriate level of knowledge while enhancing the adoption of recommended self-care practices with special emphasis on high-risk audience segments.

## Introduction

Novel-coronavirus disease is currently a global health threat and public health emergency of international concern [1]. The severe acute respiratory syndrome outbreak that was linked to coronavirus (SARS-COV) was first reported in 2003 [2]. Sixteen years later, a closely similar outbreak, which first received the name novel-SARS-COV2 was detected. The outbreak was first reported in late December 2019, when clusters of pneumonia cases of unknown etiology were found to be associated with epidemiologically linked exposure to the seafood market and untraced exposures in the city of Wuhan of China in Hubei Province [3]. It was by far the largest outbreak of atypical pneumonia since the SARS outbreak. In the initial stage of the outbreak, the total number of cases and deaths exceeded those of SARS [2]. Subsequently, the spread of the virus has shown exponential growth and spread to all continents and received a unique name by COVID-19 from the World Health Organization (WHO) [4]. On January 30, 2020, the WHO declared that COVID-19 is a pandemic disease [1,4]. As of December 2019 until May 7, 2020, the pandemics registered 3,886,2301,425 cases and 268, 908 deaths in the world. Ethiopia has become among the COVID-19 affected countries as of March 15, the date on which one imported case was first detected. On May 7, 2020, there were 191 total notified cases and 4 deaths in Ethiopia.

Similar to SARS, COVID-19 is a beta-coronavirus that can spread to humans through intermediate hosts such as bats [5]. Available evidence has shown that the virus spreads from human-to-human, mainly through respiratory droplets and body contacts [6,7]. Contact with contaminated surfaces, hands, and touching of faces-eye-nose-mouth are predominant ways to get exposed to the infected droplets. In top of this, some of the factors that aggravated the rapid spread of the virus were: first, on average, every case of COVID-19 will create up to 4 new cases (transmissibility = 4.08) [8]. Second, the average incubation period is estimated to be as short as 5.2 days, with variation among patients [9]. Third, the capability of asymptomatic transmission of the virus [8–11]. Thus, COVID-19 has become highly contagious and has reached out to more than 200 countries within 3 months.

COVID-19 has no effective cure, yet early recognition of symptoms and timely seeking of supportive care and preventive practices enhance recovery from the illness and combat the spread of the virus. The symptoms of COVID-19 infection include fever, fatigue, cough, sore throat, breathing difficulty, myalgia, nausea, vomiting, and diarrhea [12,13]. Older men with medical comorbidities are more likely to get infected, with worse outcomes. Severe cases can lead to cardiac injury, respiratory failure, acute respiratory distress syndrome, and death [12,13]. The provisional case fatality rate by WHO is approximately 3.4% [4]. Knowledge of

the symptoms, high-risk conditions, risky practices, and prognosis is of paramount importance to curb the pandemic by boosting the probability of practicing avoidance of contact with contaminated surfaces/hands/ objects, washing of hands, keeping physical distances, taking precautions while coughing/sneezing, using an alcohol-based rub and other protective equipment.

Even though there are strong initiatives and recognition of the public health importance of COVID-19 by the Ethiopian government (screening, quarantine, and treatment centers), there is a strong need to reinforce community awareness and practices to stop the nationwide spread of the virus. Therefore, this study aimed to assess knowledge, perceptions, and practices that inform communication and community engagement efforts in the fight against COVID-19 among community members who visited Jimma University Medical Center (JUMC) in Jimma town, Southwest Ethiopia.

## Methods and materials

### Study setting and period

This study was conducted in JUMC during March 22–28, 2020, i.e. within two weeks of the first COVID-19 case in Ethiopia. JUMC is located in Jimma town, the capital of the Jimma zone in Oromia National Regional State. JUMC provides specialized and referral diagnostic and treatment services. As one of the limited national COVID-19 testing centers, JUMC is readily organized to serve quarantine and treatment with 200 consecrated beds.

### Study design

A hospital-based cross-sectional study design was conducted to rapidly assess knowledge, perceptions, and practices among clients and patients who visited the medical center.

### Population and sample

The study was conducted among selected JUMC visitors of all kinds. The visitors expectedly came from different localities, including regional towns for referral services. The visitors' samples were from different wards: emergency, inpatients, outpatients, specialized clinics (maternal, TB/HIV, chronic illness, internal medicine, surgery, etc.).

### Sample size determination and sampling

The single population formula was used to determine the sample size. Accordingly, the formula for sample size determination uses is: n = $(Z_{\alpha/2})^2$ *$[(p_1q_1)/(d)^2]$, where n denotes the sample size, $Z_{\alpha/2}$ is the reliability coefficient of standard error at 5% level of significance = 1.96, p = proportion JUMC visitors who are knowledgeable about COVID-19 (50%, no previous study found), and d refers to the level of standard error tolerated (5%). Hence, the calculation yielded a sample size of 247 visitors after adjusting for the total visitors' population that was expected to visit the hospital in one-week period. An equal proportion of the sample was allocated to major wards/clinics in the hospital. Finally, consecutive sampling was applied until the allocated sample size was filled.

### Instrument and measurement

The survey was conducted using tools that were adapted from WHO resources and similar studies [14–16]. The knowledge questions had 14 items covering issues such as symptoms, risk conditions, prognosis, modes of transmission and safety, and precautions in COVID-19. Perceptions were measured by three questions about self-efficacy, collective efficacy, and stigma.

The visitors were asked about their experiences over the last few days since the onset of COVID-19 about hand washing, avoidance of shaking hands, overcrowded places, physical proximity while walking/greeting, etc. All the questions elicited a "yes/no" response. Overall knowledge status was indicated by three labels: low ($\leq 8$ of 14), moderate [9–10 of 14], and high ($\geq 11$ of 14 items).

## Data collection, management, and analysis

The data were collected through an exit interview at clinics/wards in JUMC by trained and experienced enumerators. A minimum of one-meter distance was kept between interviewers and interviewees. A pretested translated version of the instrument was used for data collection. Data analysis were managed using software for the statistical package for social science package (SPSS) version 20.0. Before further data analysis, reverse scoring for negatively worded items, normality curve, and tests of homogeneity of variances were checked. Knowledge was first analyzed item by item correct rate. Then, a multi-dimensional knowledge score (MDKS) was calculated by summing up the items. Using a quartile score, 11 of 14 correct answers were used as a cutoff value for high knowledge. $\leq 8$ of the 14 items were labeled as low knowledge. Independent sample t-test and analysis of variance (ANOVA) were done to test differences in means of MDKS by socio-demographic variables and efficacy-stigma perceptions. Predictors of COVID-19 preventive actions that were dominantly practiced were executed using logistic regression. Adjusted odds ratios were used to interpret variables in the final model of preventive practices.

## Ethical approval and considerations

The study was ethically approved by the institutional review board (IRB) of Jimma University. The ethical clearance letter reference number is IRB 00097/20. Verbal informed consent was sought from every respondent. Data collectors were observed for 14 days after the completion of the survey. The interviewers wore protective face masks. Reasonable physical distance was kept between the involved individuals during data collection. The potential risk was minimal at the time of the study. The data were collected in a private condition and kept confidential.

## Results

### Socio-demographic characteristics of the respondents

A total sample of 247 JUMC visitors were approached. The respondents' mean age was 30.5 ±10.2 years. More than 3/4th, 189 were men. 52 (21.1%) of visitors were unable to read or write. A higher proportion of respondents were married (62.3%), Muslims (59.9%), and farmers (31.2%). Table 1 presents the details of the socio-demographic characteristics.

### Knowledge and perception of COVID-19 among hospital visitors

Table 2 presents the details of knowledge about COVID-19. Analysis of quartiles of knowledge about COVID-19 revealed that >50% of JUMC visitors correctly responded to 10 of 14 knowledge items. The knowledge components are presented as follows.

**Symptoms.** 205 (83.0%) of the visitors knew the main clinical symptoms of COVID-19 as fever, fatigue, dry cough, and myalgia. In fact, < 2/5th (37.7%) of the respondents mentioned other symptoms such as the stuffy nose, runny nose, and sneezing, which distinguishes COVID-19 from common cold/flu.

**Risk factors and prognosis.** One hundred seventy-nine (72.5%) of the visitors knew that elderly people who have chronic illnesses and obesity are at higher risk of developing a severe

**Table 1. Socio-demographic characteristics of JUMC visitors, Jimma, March 2020 (n = 247).**

| Variable | Frequency (n) | % |
|---|---|---|
| **Age (years)** | | |
| ≤19 | 22 | 8.5 |
| 20–29 | 110 | 44.5 |
| 30–39 | 65 | 26.3 |
| 40–49 | 36 | 14.6 |
| 50–59 | 10 | 4.0 |
| ≥60 | 4 | 1.6 |
| *Mean (±St.D) of age (in years)* | *30.5(mean)* | *(±10.2)St.D* |
| **Sex** | | |
| Male | 189 | 76.5 |
| Female | 58 | 23.5 |
| **Educational status** | | |
| Cannot read and write | 52 | 21.1 |
| Read and write | 20 | 8.1 |
| Primary (1–8 grade) | 52 | 21.1 |
| Secondary (9–12 grade) | 62 | 25.1 |
| College and above | 61 | 24.7 |
| **Marital status** | | |
| Single | 91 | 36.8 |
| Married | 154 | 62.3 |
| Divorced | 1 | 0.4 |
| Widowed | 1 | 0.4 |
| **Occupational status** | | |
| Farmer | 77 | 31.2 |
| Student | 75 | 30.4 |
| Currently unemployed | 38 | 15.4 |
| Government employed | 34 | 13.8 |
| Private business/employed | 23 | 9.3 |
| **Religion** | | |
| Muslim | 148 | 59.9 |
| Orthodox | 56 | 22.7 |
| Protestant | 39 | 15.8 |
| Others | 4 | 1.6 |
| **Monthly Income (ETB)** | | |
| ≤499 | 122 | 49.4 |
| 500–2000 | 62 | 25.5 |
| ≥2001 | 63 | 25.1 |
| Median (mean) | 500 (1,723.2) | |

ETB = Ethiopian Birr, JUMC = Jimma Medical Center

form of COVID-19. Almost the same proportion knew that COVID-19 had no effective cure yet early seeking of treatment increases the chance of recovery.

**Mode of transmission.** A high proportion (95.1%) of the visitors knew that the COVID-19 virus spreads via respiratory droplets of infected people. However, 77 (31.2%) of the respondents reported that asymptomatic transmission is possible.

**Table 2. Knowledge of COVID-19 among JUMC visitors, Jimma, Ethiopia, March 2020 (N = 247).**

| Variable (n = 247) | Frequency (%) | |
|---|---|---|
| | Correct | Incorrect |
| **Knowledge of symptoms** | | |
| The main clinical symptoms of COVID-19 are fever, fatigue, dry cough, and myalgia | 205 (83.0) | 42(17.0) |
| Unlike the common cold, stuffy nose, runny nose, and sneezing are less common in persons infected with the COVID-19 virus | 93 (37.7) | 154(62.3) |
| **Knowledge of high risk and prognosis** | | |
| Not all persons with COVID-2019 will develop severe cases. Only those who are elderly, have chronic illnesses & are obese are more likely to be severe cases | 179 (72.5) | 68 (27.5) |
| There currently is no effective cure for COVID-2019, but early symptomatic and supportive treatment can help most patients recover from the infection | 178 (72.1) | 69 (27.9) |
| **Knowledge about Mode of transmissions and infectiousness** | | |
| The COVID-19 virus spreads via respiratory droplets of infected individuals | 235 (95.1) | 12 (4.9) |
| Eating or contacting wild animals would result in the infection by the COVID-19 virus | 119 (48.2) | 128(51.8) |
| Persons with COVID-19 cannot infect the virus to others when a fever is not present * | 77 (31.2) | 170(68.8) |
| **Knowledge about ways of prevention** | | |
| Proper washing hand with soap and water is one method of preventing COVID-19 | 236 (95.5) | 11 (4.5) |
| One way of prevention of COVID 19 is not touching the eye, nose by unwashed hands | 229 (92.7) | 18 (7.3) |
| To prevent the infection by COVID-19, individuals should avoid going to crowded places such as train stations and avoid taking public transportations | 223 (90.3) | 24 (9.7) |
| Ordinary residents can wear general medical masks to prevent the infection by the COVID-19 virus | 216 (87.4) | 31 (12.6) |
| People who have contact with someone infected with the COVID-19 virus should be immediately isolated in a proper place | 216 (87.4) | 31 (12.6) |
| Isolation and treatment of people who are infected with the COVID-19 virus are effective ways to reduce the spread of the virus | 212 (85.8) | 35 (14.2) |
| Children and young adults don't need to take measures to prevent the infection by the COVID-19 virus * | 15 (6.1) | 232 (93.9) |
| **Quartiles of correctly answered knowledge (of 14 items)** | | |
| Quartile 1 (0–25%) | 1–8 of 14 | - |
| Quartile 2 [25–50%) | 9 of 14 | - |
| Quartile 3 [50–75%) | 10 of 14 | - |
| Quartile 4 [75–100%) | 11-14of14 | - |

* Correction rate calculated from 'no' response for false statements,

** MDKS constructed from 14 correct items  JUMC = Jimma Medical Center

**Prevention practices.** Properly washing hands with soap and water (95.5%), not touching eye-nose-mouth with unwashed hands (92.7%), and avoiding crowded places (90.3%) were commonly known methods of preventing COVID-19 transmission. However, 15 (6.1%) mentioned that children and young adults must take measures to prevent infection by the COVID-19 virus.

## Multi-dimensional knowledge status of COVID-19 among hospital visitors

Multidimensional (symptoms, risk factors and prognosis, transmission modes, and preventive methods) analysis of knowledge of COVID-19 indicated that 41.7% and 41.3% of JUMC

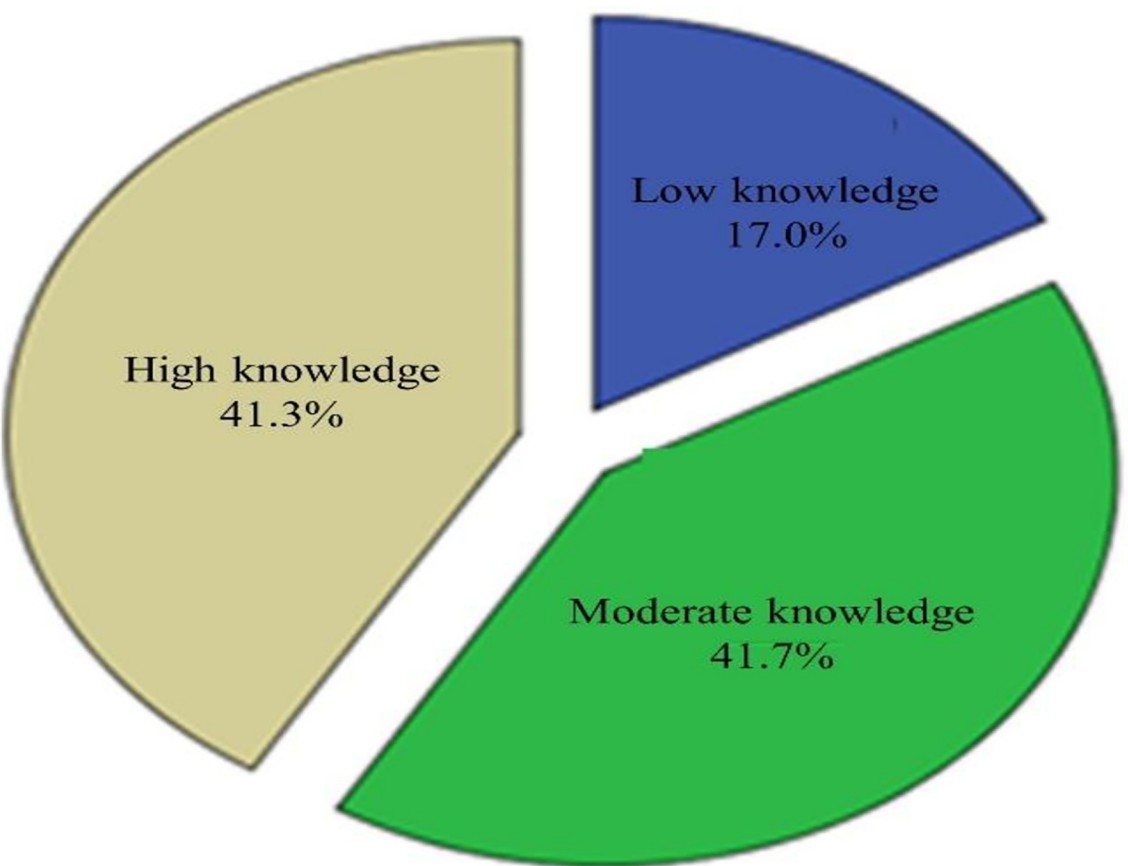

**Fig 1. Pie chart indicating multi-dimensional knowledge status about COVID-19, Jimma-Ethiopia, March, 2020.**

visitors were moderately and highly knowledgeable, respectively, (Fig 1). The line graph shows counts of correctly answered knowledge items(score ≥11) referred to as a highly knowledgeable class (Fig 2).

## Exposure to training and perception to combat the spread of COVID-19 among JUMC visitors

Only 8(3.2%) visitors reported exposure to organized educational sessions about COVID-19. Two hundred seven, 83.8% of the visitors felt that COVID-19 was a stigmatized disease, and 68.8% of the visitors perceived self-efficacy as to control (Fig 3).

## Differences in knowledge and perceptions by sociodemographic variables

Analysis of variance (ANOVA) indicated that the multidimensional knowledge (MDK) score on COVID-19 was significantly different by some socio-demographic variables (educational status, age groups, and occupation). For example, post hoc tests using Bonferroni (equal variance assumed) and Tamhane (unequal variance assumed) methods showed that visitors whose educational status was secondary school and above had a higher mean MDK score ($F = 5.38$, $p<0.01$) compared to lower graders and non-attendants of formal education. Additionally, visitors whose ages ranged between 30 and 49 years had lower mean MDK scores compared to younger visitors ($F = 2.29$, $p<0.05$). Farmer visitors had lower mean MDK scores compared to

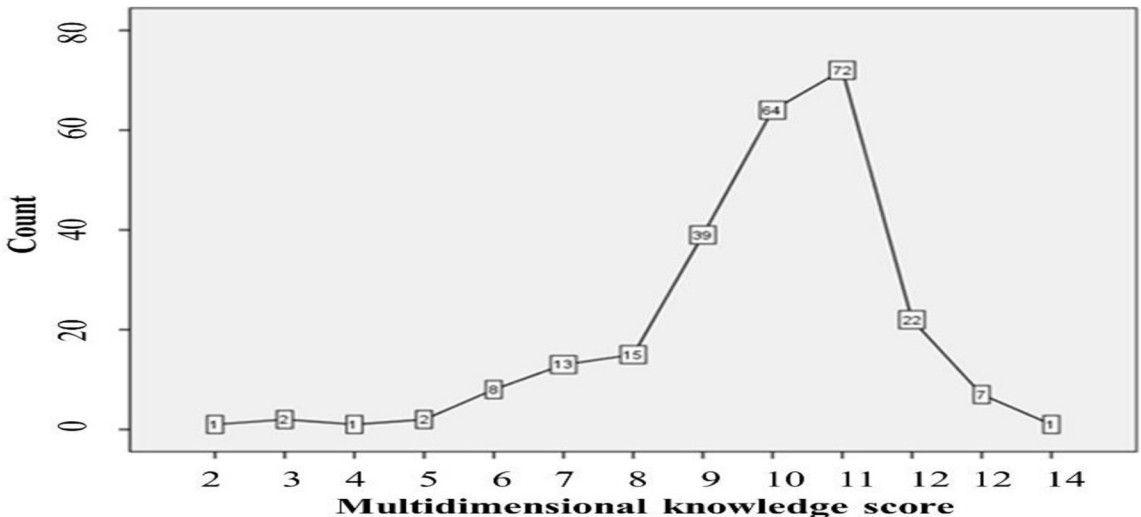

**Fig 2. Line graph showing counts for summation scores of knowledge items about COVID-19, Jimma-Ethiopia, March, 2020.**

employment in private/government businesses and students. The unemployed ones had lower mean compared to farmers (F = 5.51, p<0.05). Moreover, an independent sample t-test showed that unmarried visitors had a higher mean MDK score (t = 2.64, p<0.05) compared to married visitors. Respondents' religious affiliations, sex, and perceived self-efficacy showed no significant differences in means of MDK score (F = 1.25, p>0.25), (t = 1.74, p>0.25), and (t = 1.92, p>0.05), respectively. The perception of self-efficacy to combat COVID-19 showed no significant difference by socio-demographic variables.

## COVID-19 preventive practices among JUMC visitors

Over the last few days, JUMC visitors were predominantly engaged in frequent hand washing with water and soap (77.3%), stopped shaking hands while giving greeting (53.8%), avoiding physical proximity (33.6%) and going to crowed places (33.2%), to protect themselves from COVID-19. Table 3 presents details.

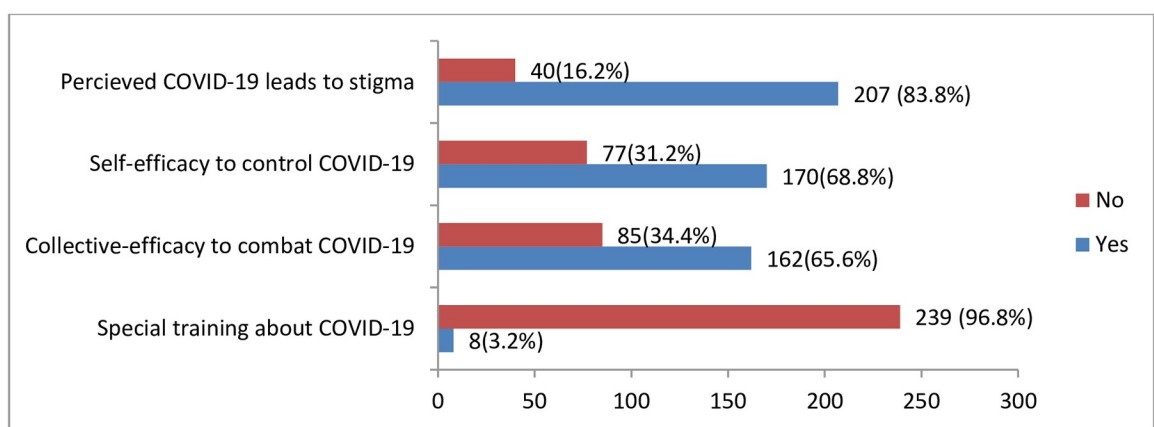

**Fig 3. Bar graph including experience of training session and perceived stigma and efficacy to combat COVID-19, JMC visitors, Jimma-Ethiopia, March, 2020.**

**Table 3. JUMC visitors COVID-19 preventive practices, Jimma, Ethiopia, March 2020 (n = 247).**

| Practice variables | Frequency (%) | |
|---|---|---|
| Over the last few days following the report of COVID-19 in Ethiopia, I… | Yes (%) | No (%) |
| am frequently washing hands with water and soap | 191 (77.3) | 56 (22.7) |
| stopped shaking hands while giving greeting | 133 (53.8) | 114 (46.2) |
| avoided proximity including while greeting (within 1 meter) | 83 (33.6) | 164 (66.4) |
| have not been going to crowed place | 82 (33.2) | 165 (66.8) |
| wore a mask when leaving home | 35 (14.2) | 212 (85.8) |
| avoid touching eye, nose, mouth before washing hands | 28 (11.3) | 219 (88.7) |
| used cover /elbow for coughing/sneezing | 28 (11.3) | 219 (88.7) |
| others (alcohol-rubbing, no contact with surfaces) | 14 (5.7) | 233 (94.3) |
| Have started to stay home | 4 (1.6) | 243 (98.4) |

## Predictors of engagement in COVID-19 major preventive behaviors

Socio-demographic characteristics, knowledge, and self-efficacy were important factors that predicted the adaptation of measures that protect from COVID-19. Table 4 provides the details.

### Predictors of frequent hand-washing practice

JUMC visitors who were—40–49 years old and unemployed were averagely 11.1 and 3.6 times less frequently washed their hands over the last few days compared to their counterparts in youngest age groups and farming occupations, respectively. Visitors who belonged to the highest knowledge class were averagely 3.48 times washers compared to those with low knowledge. The majority of specific knowledge and perceived efficacy items had a crude and positive effect on the practice of washing hands. Overall, the above predictors explained 38.7% of the variance in frequent hand washing practices.

### Predictors of avoidance of handshaking practice

JUMC visitors who were employed at private business and government offices were averagely 5.7 and 2.68 times more likely to avoid the practices of shaking hands for greeting over the last few days compared to those whose occupation was farming, respectively. Concerning specific knowledge predictors, respondents who perceived animal contacts spread COVID-19 and wearing masks prevented the infection by COVID-19 were 2.73 and 2.88 times more likely careful to avoid shaking hands. Visitors who belonged to the highest knowledge class were on average 2.45 times avoiders of shaking hands compared to those with low knowledge. Visitors who felt self-efficacious to successfully control COVID-19 were 3.89 times more likely to engage in avoidance of shaking hands compared to those with low efficacy. The crudes odds ratio showed that many of the knowledge items and socio-demographic variables (being females, higher educational levels) have positively influenced the avoidance of shaking hands. Overall, the above predictors explained 32.9% of the variance in avoiding hand shaking practice.

## Discussion

This study assessed the knowledge, perceptions, and practices of COVID-19 following the onset of the pandemics in Ethiopia. Thus, it would inform public education and engagement to stop the spread of this contagious virus before it is too late. Despite a significant proportion

**Table 4. Predictors of COVID-19 preventive measures, JUMC visitors, Jimma, March 2020 (n = 247).**

| Factor variables | Handwashing with soap & water | | Stopped shaking hands | |
|---|---|---|---|---|
| **Socio-demographic** | **COR (95% CI)** | **AOR** | **COR** | **AOR** |
| **Age groups** | | | | |
| < = 19 | 1 | 1 | 1 | 1 |
| 20–29 | 1.00 (0.26,3.80) | 0.78(0.14,4.25) | 2.34(0.92,5.94) | 1.54(0.45,5.22) |
| 30–39 | 2.24(0.60,8.54) | 0.29(0.04,2.21) | 1.58(0.60,4.22) | 1.16(0.25,5.29) |
| 40–49 | 0.16(0.04,0.63)* | 0.09(0.01,0.74)* | 1.03(0.35,3.03) | 1.90(0.30,12.32) |
| > = 50 | 1.73(0.30,10.10) | 0.42(0.04,4.38) | 1.44(0.38,5.57) | 1.61(0.75,3.44) |
| **Sex** | | | | |
| Male | 1 | 1 | 1 | 1 |
| Female | 1.02(0.50,2.01) | 0.76(0.27,2.12) | 1.88(1.02,3.47)* | 1.61(0.75,3.44) |
| **Educational status** | | | | |
| Neither read nor write | 0.26(0.10,0.67)* | 0.28(0.04,1.76) | 0.26(0.12,0.57)* | 0.64 (0.16,2.55) |
| Read and write | 0.35(0.11,1.18) | 0.33(0.04,2.66) | 0.91(0.31,2.62) | 2.55(0.50,13.10) |
| Primary school | 0.45(0.17,1.20) | 0.34(0.07,1.60) | 0.42(0.20,0.90)* | 0.79(0.25,2.49) |
| Secondary school | 0.79(0.29,2.15) | 0.35(0.09,1.50) | 0.72(0.35,1.51) | 0.81(0.30,2.19) |
| College and above | 1 | 1 | 1 | 1 |
| **Religious affiliation** | | | | |
| Muslim | 1 | 1 | 1 | 1 |
| Orthodox | 1.59(0.73,3.46) | 0.70(0.26,1.95) | 1.77(0.95,3.31) | 1.03(0.47,2.23) |
| Protestant | 1.34(0.57,3.16) | 1.12(0.32,4.00) | 2.29(1.10,4.80)* | 2.02(0.78,5.20) |
| **Marital status** | | | | |
| Single | 1 | 1 | 1 | 1 |
| Married | 029(0.14,0.61)* | 0.44(0.13,1.45) | 0.66(0.39,1.11) | 0.70(0.28.1.75) |
| **Occupation** | | | | |
| Farmer | 1 | 1 | 1 | 1 |
| Student | 3.12(1.33,7.31)* | 0.30(0.05,1.75) | 1.89(0.99,3.60) | 1.15(0.34,3.89) |
| Unemployed | 0.73(0.32,1.66) | 0.28(0.08,0.97)* | 1.34(0.61,2.92) | 1.60(0.57,4.46) |
| Private employed | 2.02(0.62,6.61) | 1.08(0.18,6.34) | 4.20(1.49,11.8)* | 5.70(1.42,22.7)* |
| Gov't employed | 1.99(0.73,5.45) | 0.34(0.05,2.25) | 4.12(1.70,10.0)* | 2.68(1.01,7.12)* |
| **Knowledge (yes)** | | | | |
| Knew main clin.symptom | 3.75(1.86,7.59)* | 3.30(1.03,19.54)* | 2.77(1.38,5.57)* | 1.62(0.5,4.99) |
| Differentiated COVID-19 from symptoms of flu | 2.38(1.20,4.72)* | 1.64(0.58,4.59) | 2.17(1.28,3.70)* | 1.29(0.51,3.24) |
| Knew high risk group | 2.04(1.10,3.83)* | 1.10(0.38,3.18) | 1.72(0.98,3.01) | 0.75(0.2,2.03) |
| Early supportive treatments increase recovery | 2.70(1.44,5.04) * | 1.23(0.45,3.41) | 2.50(1.41,4.42)* | 1.75(0.66,4.66) |
| COVID-19 spreads via respiratory droplets | 3.70(1.14,11.97)* | 1.72(0.30,9.92) | 2.43(0.71,8.31) | 1.59(0.28,9.05) |
| Eating or contact with animals spreads COVID-19 | 1.93(1.10,3.54)* | 1.11(0.44,2.79) | 1.94(1.17,3.22)* | 2.73(1.27,5.88)* |
| Proper hand washing prevents COVID-19 | 4.46(1.31,15.23)* | 2.05(0.30,13.97) | 3.27(0.85,12.63) | 1.49(0.19,11.94) |
| Wearing mask prevent infection by COVID-19 | 1.76(0.78,4.00) | 0.96(0.30,3.21) | 3.30(1.45,7.49)* | 2.88(1.04,8.00)* |
| Avoiding crowed place prevents COVID-19 | 4.95(2.10,11.80)* | 3.80(1.08,13.34)* | 1.7(0.73,4.04) | 1.07(0.28,4.11) |
| Isolation of infected people prevents COVID-19 | 2.70(1.27,5.75)* | 1.52(0.47,4.96) | 0.98(0.48,2.00) | 0.64(0.20,2.01) |
| Immediate contact isolation prevents COVID-19 | 4.00(1.83,8.75)* | 3.89(1.24,12.17)* | 1.28(0.60,2.73) | 1.03(0.34,3.12) |
| **MDK status** | | | | |
| Less | 1 | 1 | 1 | 1 |
| Moderate | 2.01(0.94.4.31) | 1.98(0.82,4.82) | 2.12(1.00,4.48)* | 1.86(0.82,4.25) |
| High | 4.66(2.00,10.87)* | 3.48(1.34,9.09) * | 3.67(1.72,7.84)* | 2.45(1.06,5.65) * |
| **Attitude & efficacy (yes)** | | | | |
| Self-efficacy to control COVID-19 | 3.54(1.90,6.57)* | 1.58(0.45,5.61) | 2.83(1.62,4.94)* | 3.89(1.44,10.50)* |

*(Continued)*

**Table 4.** (Continued)

| Factor variables | Handwashing with soap & water | | Stopped shaking hands | |
|---|---|---|---|---|
| Socio-demographic | COR (95% CI) | AOR | COR | AOR |
| Collective efficacy to control COVID-19 | 2.85(1.55,5.26)* | 2.99(0.83,10.72) | 1.75(1.03,2.98)* | 0.78(0.30,2.05) |

R-square ($R^2$) = 38.7% and 32.9% for hand washing and not-shaking hands,

*significant at $p < 0.05$,

COR: crude odds ratio, AOR: adjusted odds ratio

(83.0%) of JUMC visitors knew the main clinical symptoms of COVID-19, only 37.7% of them were able to identify symptoms like (stuffy nose, runny nose, and sneezing) that distinguish it from common cold/flu. Confusing the symptoms of COVID-19 with a common cold could challenge the practice of early treatment-seeking when COVID-19 is accompanied with flu/common cold-like symptoms. Moreover, confusion can intensify bias and social stigma related to COVID-19. According to the present study, 83.8% of the visitors perceived that COVID-19 is a disease leading to social stigma. In history, similar epidemics have been associated with stigma [17,18]. The WHO recommends that any communication and case management efforts should simultaneously address the stigma associated with COVID-19 by providing do's and don'ts [19]. This indicates that governments, media, health facilities, and local organizations are expected to integrate stigma reduction interventions across all COVID-19 combating activities.

Concerning knowledge of risk factors and prognosis, good proportions (72.5%) of the JUMC visitors knew that elderly people who have chronic illnesses and obesity are at high risk of developing a severe form of COVID-19. Moreover, 72.1% knew that early seeking of treatment increases the chance of recovery, despite the lack of effective cure developed till the moment. A study in Wuhan, China revealed that deaths are can reach up to 16.7% (9.4–23.9%) for patients in an intensive care unit [20, 21]. The provisional case fatality rate by WHO is approximately 3.4% [4]. Although this level of knowledge is promising to safeguarding the most at-risk population segment, it could divert the attention of the young/adult public to adopt preventive measures. The misperception that young/children are at low risk of COVID-19 is found under the WHO's list of rumors that could grant false assurances [14, 22]. The findings from the current study support this idea, as only 15 (6.1%) of the respondents perceived that children and young adults must take measures to prevent infection by the COVID-19 virus. This requires adequately addressing children and young people in COVID-19 communication messaging.

This study found out that the knowledge about the major mode of transmission was as high as 95.1% i.e. the visitors knew that COVID-19 virus spreads via contamination with respiratory droplets of infected people. Although the current study did not investigate potential sources where infections can be acquired, a study conducted in Wuhan, China, presented 10(9.8%) of human-to-human transmission of COVID-19 as familial clustering of cases, and 34(33.3%) nosocomial infections [23]. In this study, only 77 (31.2%) of the respondents knew that asymptomatic people who do not even present with fever can still transmit the virus. The latter implies two main points. First, there could be a compromise in the adaptation of safety precautions measures, especially the practice of covering the breathing system during coughing/sneezing and proximity one can have with potentially infected ones. Second, in resource-limited settings where nationwide active searching for infected people is minimal or absent, the chance of asymptomatic transmission and spread of the novel-coronavirus would be high [3,10,22]. In this regard, Ethiopia has not yet decentralized diagnostic centers very well.

Concerning the knowledge about methods of preventing infection by COVID-19, JUMC visitors dominantly mentioned proper washing of hands (95.5%), not touching face-eye-nose-mouth before washing hands (92.7%), and avoiding crowded places (90.3%). If the above knowledge is executed in self-care practices by individuals and the public at large, it can help prevent the spread of the virus in the country.

Nevertheless, there were huge gaps between the magnitude of knowledge of preventive methods and the practices. For example, only 77.3% of visitors had reportedly frequently washed their hands with water and soap, although the knowledge was as high as 95.5%. In the same manner, those who reported that they had not been going to crowded places were 33.2%, although the knowledge that avoidance of crowded places prevents infection by novel-coronavirus was 90.3%. Perhaps, in resource-limited settings, several reasons can be mentioned as to why people cannot frequently and properly wash their hands and easily avoid crowded places. For instance, to mention some, Ethiopia is known by modest coverage and intermittent continuity of water supply and hand washing facilities, a high rate of overcrowded living conditions, frequent social and religious ceremonies, and high unemployment rate calling for urgent efforts to bridge the gap between knowledge and practices [24, 25].

This study found out that hand washing and avoidance of shaking hands for greeting as two dominantly practiced methods of preventing infection by COVID-19. Still, avoidance of non-careful touching of face parts that are used for the entrance of the virus, wearing masks/cover while coughing/sneezing, use of sanitizers, and stay at home was very low (1.6–11.3%). The entire practices were not as high in contrast to the contagiousness natures of the virus. In the absence of adaptation to these practices as packages, the entire public is at high to contract the infection [8–10].

Analysis of predictors of frequent hand washing (with soap and water) showed that older age groups, particularly in 40–49 years and unemployed, were risk social groups. The above discussions about social variables similarly applicable here. Highly knowledgeable visitors were averagely more than threefold more likely to frequently wash their hands compared to less knowledgeable visitors. Specifically, knowledge of the main clinical symptoms, the necessity of avoiding overcrowded places, and immediately isolating the contacts with infected people as a means of prevention were all more than threefold more likely to go for washing hands frequently as a fight against COVID-19. Accurate, timely, and relevant knowledge is consistently associated with a mild threat that enhances the adaptation of safety practices and precautions [16, 17].

Similarly, working for private and government businesses were positive predictors of avoidance of shaking hands for greeting. Perhaps, they had strict instructions about avoidance of shaking hands at offices compared to farmers. Highly knowledgeable visitors were averagely more than twofold more likely to avoid shaking hands compared to less knowledgeable visitors. Specifically, visitors who were more careful about eating/contacts with animals, leaning towards wearing masks to prevent COVID-19, and felt self-efficacious to successfully control COVID-19 were more than two-threefold more likely to avoid handshaking to combat COVID-19. It looked these people were more concerned about contacts and adapted with hygienic precautions.

Finally, the authors would like to report the limitations of the study in that the findings were not well discussed in the related literature. To the best of our knowledge, there have been no similar published studies in Ethiopia. Moreover, this study relied on self-reported practices. In this urgent time of dealing with pandemic, there would reach a certain level of social desirability that can bias the reports about the practices.

## Conclusions

The magnitude of the visitors' knowledge, perceived self-efficacy to control, and preventive practices such as hand washing, avoidance of handshaking, and physical distancing were modest to protect themselves from this highly contagious virus. Notably, some knowledge/perception about COVID-19 were very low and needs an urgent improvement: "*children/young adults must take measures to prevent the infection (6.1%)*", "*asymptomatic-in the absence of fever transmission (31.2%)*" and "s*ymptoms that distinguished it from the common cold (37.7%)*". There were huge discrepancies between knowledge of prevention methods and actual practices, especially hand washing. In such highly contagious viruses with no effective cure, high-level knowledge must be achieved in the population to stop the spread of the virus. High-level knowledge about the disease and its prevention methods predicted the practice of frequent hand washing and avoidance of hand-shaking. Perceived personal and collective efficacy to stop COVID-19 enhanced dominant practices. Therefore, risk communication and community engagement efforts for combat COVID-19 should emphasize addressing key preventive methods, use dissonant messages to close the gaps between existing knowledge and actual behaviors, and keep advancing knowledge status based on the contexts of significant socio-demographic characteristics for designing effective and tailored communication strategies. The finding also suggests interventions on COVID-19 should simultaneously address the issue of social stigma and discrimination before it gets out of control.

## Supporting information

**S1 Data. Information about pre-testing of the survey questionnaire.**
(DOCX)

**S2 Data. Knowledge, attitudes, and practices towards COVID-19 among visitors of JUMC: Cross-sectional survey.**
(DOCX)

**S3 Data. Beekumsa, Ilaalcha, gochaa wa'ee dhibee Koronaa (COVID-19) dawwatoota hospital JUMC: Qorannoo yeroo yeroo tokkoofi geggeefamu.**
(DOCX)

## Acknowledgments

We express our heartfelt thanks to all individuals who participated in the study: respondents, data collectors, and administrative officials.

## Author Contributions

**Conceptualization:** Yohannes Kebede, Yimenu Yitayih, Zewdie Birhanu, Seblework Mekonen, Argaw Ambelu.

**Data curation:** Yohannes Kebede, Yimenu Yitayih, Zewdie Birhanu, Argaw Ambelu.

**Formal analysis:** Yohannes Kebede.

**Funding acquisition:** Seblework Mekonen, Argaw Ambelu.

**Investigation:** Yimenu Yitayih, Seblework Mekonen, Argaw Ambelu.

**Methodology:** Yohannes Kebede, Argaw Ambelu.

**Project administration:** Yimenu Yitayih, Seblework Mekonen, Argaw Ambelu.

**Resources:** Argaw Ambelu.

**Software:** Yohannes Kebede.

**Supervision:** Yimenu Yitayih, Zewdie Birhanu, Argaw Ambelu.

**Validation:** Yohannes Kebede, Zewdie Birhanu, Seblework Mekonen, Argaw Ambelu.

**Visualization:** Yimenu Yitayih, Zewdie Birhanu, Seblework Mekonen, Argaw Ambelu.

**Writing – original draft:** Yohannes Kebede.

**Writing – review & editing:** Yohannes Kebede, Yimenu Yitayih, Zewdie Birhanu, Seblework Mekonen, Argaw Ambelu.

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
