## [Decision Letter · Decision Letter 0]

6 May 2020

PONE-D-20-11944

Knowledge, perceptions and preventive practices towards COVID-19 among Jimma University Medical Center visitors, Southwest Ethiopia

PLOS ONE

Dear Mr. Kebede,

Thank you for submitting your manuscript to PLOS ONE. After careful consideration, we feel that it has merit but does not fully meet PLOS ONE’s publication criteria as it currently stands. Therefore, we invite you to submit a revised version of the manuscript that addresses the points raised during the review process.

We would appreciate receiving your revised manuscript by Jun 20 2020 11:59PM. To enhance the reproducibility of your results, we recommend that if applicable you deposit your laboratory protocols in protocols.io, where a protocol can be assigned its own identifier (DOI) such that it can be cited independently in the future. For instructions see: http://journals.plos.org/plosone/s/submission-guidelines#loc-laboratory-protocols

We look forward to receiving your revised manuscript.

Kind regards,

Wen-Jun Tu

Academic Editor

PLOS ONE

Journal Requirements:

1. Please include additional information regarding the survey or questionnaire used in the study and ensure that you have provided sufficient details that others could replicate the analyses. For instance, if you developed a questionnaire as part of this study and it is not under a copyright more restrictive than CC-BY, please include a copy, in both the original language and English, as Supporting Information. Moreover, please include more details on how the questionnaire was pre-tested, and whether it was validated.

3. Please ensure that you refer to Figure 3 in your text as, if accepted, production will need this reference to link the reader to the figure.

Reviewers' comments:

Reviewer's Responses to Questions

**Comments to the Author**

1. Is the manuscript technically sound, and do the data support the conclusions?

Reviewer #1: Yes

2. Has the statistical analysis been performed appropriately and rigorously? 

Reviewer #1: Yes

3. Have the authors made all data underlying the findings in their manuscript fully available?

Reviewer #1: Yes

4. Is the manuscript presented in an intelligible fashion and written in standard English?

Reviewer #1: Yes

5. Review Comments to the Author

Reviewer #1: The authors studied the knowledge, perceptions, and practices among Jimma University medical center (JUMC) visitors in Jimma town. They analysed 247 sampled visitors, 83% knew the main clinical symptoms of COVID-19, only 6.1% knew that children and young adults have to involve preventive measures,they concluded the status of knowledge and desirable practices were not sufficient enough to combat this rapidly spreading virus. COVID-19 risk communication and public education efforts should focus on building appropriate level of knowledge while enhancing adoption of recommended self-care practices with special emphasize on high-risk audience segments. I will give some comments as followings.

The sampled visitors was mostly young people between 20-29, and the female persons just conclude 58, this may lead to deviation, so the authors can explain the possible divation?

What precautions have been taken locally to share especially in Ethiopia?

3. The style of references are no consistent, please edit it.

4.The following references should be discussed in the revision text.

Cao JL, Hu XR, Tu WJ., & Liu Q. (2020). Clinical Features and Short-term Outcomes of 18 Patients with Corona Virus Disease 2019 in Intensive Care Unit. Intensive Care Medicine, DOI: 10.1007/s00134-020- 05987-7.

Cao JL, Tu WJ, Hu XR, & Liu Q. (2020). Clinical Features and Short-term Outcomes of 102 Patients with Corona Virus Disease 2019 in Wuhan,China. Clinical Infectious Diseases,DOI: 10.1093/cid/ciaa243/ 5814897.

6. PLOS authors have the option to publish the peer review history of their article (what does this mean?). If published, this will include your full peer review and any attached files.

Reviewer #1: No

---

## [Author Response · Author response to Decision Letter 0]

7 May 2020

Rebuttal letter 

Response to reviewers

We thank the editor and reviewers for evaluating our manuscript. We provide line by line responses to comments raised. Please, follow our responses in a yellow mark to every comment/question in green mark. We copied the review decision from submission menu of the editorial manager. We used the same letter to write our responses. 

PONE-D-20-11944

Knowledge, perceptions and preventive practices towards COVID-19 among Jimma University Medical Center visitors, Southwest Ethiopia

PLOS ONE

Dear Mr. Kebede,

Thank you for submitting your manuscript to PLOS ONE. After careful consideration, we feel that it has merit but does not fully meet PLOS ONE’s publication criteria as it currently stands. Therefore, we invite you to submit a revised version of the manuscript that addresses the points raised during the review process.

We would appreciate receiving your revised manuscript by Jun 20 2020 11:59PM. To enhance the reproducibility of your results, we recommend that if applicable you deposit your laboratory protocols in protocols.io, where a protocol can be assigned its own identifier (DOI) such that it can be cited independently in the future. For instructions see: http://journals.plos.org/plosone/s/submission-guidelines#loc-laboratory-protocols

• A rebuttal letter that responds to each point raised by the academic editor and reviewer(s). This letter should be uploaded as separate file and labeled 'Response to Reviewers'.

• A marked-up copy of your manuscript that highlights changes made to the original version. This file should be uploaded as separate file and labeled 'Revised Manuscript with Track Changes'.

• An unmarked version of your revised paper without tracked changes. This file should be uploaded as separate file and labeled 'Manuscript'.

We look forward to receiving your revised manuscript.

Kind regards,

Wen-Jun Tu

Academic Editor

PLOS ONE

Journal Requirements:

Comment: 

1. Please include additional information regarding the survey or questionnaire used in the study and ensure that you have provided sufficient details that others could replicate the analyses. For instance, if you developed a questionnaire as part of this study and it is not under a copyright more restrictive than CC-BY, please include a copy, in both the original language and English, as Supporting Information. Moreover, please include more details on how the questionnaire was pre-tested, and whether it was validated.

Responses: We thank you, indeed for the advices on the revision. We have attached the questionnaires we used for the current survey. Please, check in the supplementary files. Given that this study was done urgently to inform public education and risk communication effort, we pre-tested the material on very limited people (12 hospital visitors=5% of samples). Moreover, because the questionnaire was adapted from WHO resources and other similar studies in China, we did not follow standard validation procedure than pre-testing. Please find few points in supplementary files as to how pretesting was going. 

Comment: 

Response: We thank you for advising us to make editions to the writing in English regarding this manuscript. We used “grammarly” online software to edit the spelling, grammar and language usage. Following that grammarly-online suggested there are word choice issues we approached PubSURE a research paper language editing service. Accordingly, there was 227 issues (with quality score of 70%=boarder line good) found on the draft manuscript. Then taking the comment reported by PubSure we cleared most of the issues, left some that were illogical. Later, in the final revised draft we got the 196/227 issues resolved. Moreover, we used a PubSURE software and we secured a certificate of quality of language that has pass score of 97%. For your information we have attached both certificates as other additional attachment with file name “Other files”. Regarding use of professional editing, we don’t afford to pay the required payment. Moreover, currently we have made changes in the manuscript in track changes and uploaded as supporting files. We uploaded a clean copy of the edited manuscript with file name *manuscript* file

Comment: 

3. Please ensure that you refer to Figure 3 in your text as, if accepted, production will need this reference to link the reader to the figure.

Response: Yes, we thank you for advising to suggest where in the manuscript figure 3 should refer to. Accordingly, you can please refer to line 184 that we now refer figure in bracket at the end of the paragraph as such, “(Figure 3)”. 

Reviewers' comments:

Reviewer's Responses to Questions

Comments to the Author

1. Is the manuscript technically sound, and do the data support the conclusions?

Reviewer #1: Yes

2. Has the statistical analysis been performed appropriately and rigorously?

Reviewer #1: Yes

3. Have the authors made all data underlying the findings in their manuscript fully available?

Reviewer #1: Yes

4. Is the manuscript presented in an intelligible fashion and written in standard English?

Reviewer #1: Yes

5. Review Comments to the Author

Comment

Reviewer #1: The authors studied the knowledge, perceptions, and practices among Jimma University medical center (JUMC) visitors in Jimma town. They analysed 247 sampled visitors, 83% knew the main clinical symptoms of COVID-19, only 6.1% knew that children and young adults have to involve preventive measures,they concluded the status of knowledge and desirable practices were not sufficient enough to combat this rapidly spreading virus. COVID-19 risk communication and public education efforts should focus on building appropriate level of knowledge while enhancing adoption of recommended self-care practices with special emphasize on high-risk audience segments. I will give some comments as followings.

The sampled visitors was mostly young people between 20-29, and the female persons just conclude 58, this may lead to deviation, so the authors can explain the possible divation?

What precautions have been taken locally to share especially in Ethiopia?

Response: Thank you for the insight about more samples was in age groups of 20-29 years. In the first place, this is all about data. But, it is true that we noted some deviations. The following explanations can be referred to about the differences. Obviously, 40% of Ethiopian population is young people whose ages are less than 30 years old. The current study revealed 44.5% of respondents belong to this age group. Regarding, female participants the following are potential reasons for relatively low % of them sampled in this study. 1- JUMC is a specialized and referral hospital (details are mentioned about JUMC in the study setting in methods section) where many people come for services from different regions of the country, especially from Southern and Southwest region. Additionally, people come from rural settings for the services. Accordingly, it is expected that more samples can include males than females- because, in Ethiopia men are often supposed to accompany referred patients than. In Ethiopia females are supposed to stay at home in such cases because they are more important to care for children and other family members who remained at home, Moreover, obviously, sickness/injury/accidents/ are is relatively common in males and females, that means, there would be more chance for males to visit hospitals than females. In fact, the ratio of female-male is nearly 50-50 in Ethiopia. Regarding deviations, as noted in the analysis section, the effects of age and gender were already reported in ANOVA and predictors’ analysis sections. Please, refer to those sections.

Comment: 

3.The style of references are no consistent, please edit it.

Response: Thank you. The reference styles are now made consistent, references # 14, 19,22, 24, and 25 (in the revised manuscript) were made consistent with others. 

Comment: 

4.The following references should be discussed in the revision text.

Cao JL, Hu XR, Tu WJ., & Liu Q. (2020). Clinical Features and Short-term Outcomes of 18 Patients with Corona Virus Disease 2019 in Intensive Care Unit. Intensive Care Medicine, DOI: 10.1007/s00134-020- 05987-7.

Cao JL, Tu WJ, Hu XR, & Liu Q. (2020). Clinical Features and Short-term Outcomes of 102 Patients with Corona Virus Disease 2019 in Wuhan,China. Clinical Infectious Diseases, DOI: 10.1093/cid/ciaa243/ 5814897.

Response: Thank you for providing these references we used them in discussion section, about knowledge of symptoms. and, included in the reference section. Please refer statements on 263-264 and 275-277 in discussion section of revised manuscript with track changes. 

6. PLOS authors have the option to publish the peer review history of their article (what does this mean?). If published, this will include your full peer review and any attached files.

Do you want your identity to be public for this peer review? For information about this choice, including consent withdrawal, please see our Privacy Policy.

Reviewer #1: No

Response: We have registered with PACE and all the figures do fit with PLOS. We downloaded from the PACE and uploaded as Figure 1, Figure 2 and Figures in RTF format.

---

## [Decision Letter · Decision Letter 1]

13 May 2020

Knowledge, perceptions and preventive practices towards COVID-19 early in the outbreak among Jimma University Medical Center visitors, Southwest Ethiopia

PONE-D-20-11944R1

Dear Dr. Kebede,

We are pleased to inform you that your manuscript has been judged scientifically suitable for publication and will be formally accepted for publication once it complies with all outstanding technical requirements.

With kind regards,

Wen-Jun Tu

Academic Editor

PLOS ONE

Additional Editor Comments (optional):

Reviewers' comments:

Reviewer's Responses to Questions

**Comments to the Author**

1. If the authors have adequately addressed your comments raised in a previous round of review and you feel that this manuscript is now acceptable for publication, you may indicate that here to bypass the “Comments to the Author” section, enter your conflict of interest statement in the “Confidential to Editor” section, and submit your "Accept" recommendation.

Reviewer #1: All comments have been addressed

2. Is the manuscript technically sound, and do the data support the conclusions?

Reviewer #1: Partly

3. Has the statistical analysis been performed appropriately and rigorously? 

Reviewer #1: Yes

4. Have the authors made all data underlying the findings in their manuscript fully available?

Reviewer #1: Yes

5. Is the manuscript presented in an intelligible fashion and written in standard English?

Reviewer #1: Yes

6. Review Comments to the Author

Reviewer #1: Dear authors, thanks for your investigation and article for Ethiopia, COVID-19 needs all of us to do our best to treat it. Hope you and your family good.

7. PLOS authors have the option to publish the peer review history of their article (what does this mean?). If published, this will include your full peer review and any attached files.

Reviewer #1: No

---

## [Editor Report · Acceptance letter]

15 May 2020

PONE-D-20-11944R1 

Knowledge, perceptions and preventive practices towards COVID-19 early in the outbreak among Jimma University Medical Center visitors, Southwest Ethiopia 

Dear Dr. Kebede:

I am pleased to inform you that your manuscript has been deemed suitable for publication in PLOS ONE. Congratulations! Your manuscript is now with our production department. 

With kind regards,

on behalf of

Dr. Wen-Jun Tu 

Academic Editor

PLOS ONE